# New Records of the Psychrophilic *Tetracladium* Species Isolated from Freshwater Environments in Korea

**Jaeduk Goh** [1] [iD]**, Yu Jeong Jeon** [1,2] **and Hye Yeon Mun** [1,*]

1 Fungal Resources Research Division, Freshwater Bioresources Research Department, Nakdonggang National Institute of Biological Resources, 137 Donam 2-gil, Sangju-si 37242, Korea

2 Wildlife Disease Research Team, National Institute of Wildlife Disease Control and Prevention(NIWDC), Ministry of Environment, 1 Songam-gil, Gwangsan-gu, Gwangju 62407, Korea

* Correspondence: outcastm@nnibr.re.kr

**Abstract:** We investigated the diversity of aquatic fungi from freshwater environments in Korea. From the survey of aquatic fungi, three *Tetracladium* (Leotiomycetes, Helotiales) strains, NNIBRFG814, NNIBRFG1997, and NNIBRFG3891, were isolated from decaying leaves of *Fraxinus rhynchophylla* located in Taebaek-si, Gangwon-do; freshwater foam in the freshwater of the Gam stream in Gimcheon-si; and from filtered freshwater of the Yangsan stream in Mungyeong-si, Gyeongsangbuk-do, respectively. The isolated fungal strains were identified using ITS sequence analysis and morphological characters. A sequence analysis indicated that the fungal isolates were closely related to *Tetracladium breve* MK371730 (100%, NNIBRFG814), *Tetracladium setigerum* KU519120 (98.96%, NNIBRFG1997), and *T. marchalianum* MK353126 (98.14%, NNIBRFG3891). In terms of growth conditions, it was confirmed that NNIBRFG814 grew best at 20 °C, did not grow above 30 °C, and grew at 5 °C as a psychrophilic fungus. NNIBRFG814 was included in the *T. breve* clade but its morphology was not similar to that of *T. breve*. Thus, NNIBRFG814 is a new species, and *T. fraxineum* sp. nov. NNIBRFG1997 and NNIBRFG3891 are the first records in Korea of *T. setigerum* and *T. marchalianum*, respectively.

**Keywords:** aquatic hypomycetes; *Tetracladium*; *T. fraxineum*; *T. setigerum*; *T. marchalianum*; freshwater; psychrophilic fungi





## 1. Introduction

Aquatic hyphomycetes are common fungi that play key roles in the decomposition of plant matter in freshwater [1]. Ingold was the first to describe them, documenting a collection of new fungal species with aquatic life cycles [2]. The classification and identification of aquatic hyphomycetes are based on their morphology and development, and, in particular, on conidia (asexual spores) [3]. This class of fungi predominantly produces branched multiradiate or sigmoid conidia, which can easily attach to leaves or other smooth surfaces [4]. Although many aquatic hyphomycetes share morphological characters, they are not a monophyletic group, but belong to a polyphyletic taxon [5]. In a previous study in Korea, six aquatic hyphomycetes were isolated from plant litter [6,7].

*Tetracladium* De Wild. (Leotiomycetes, Helotiales) was the first aquatic hyphomycete genus to be described [2]. To date, eleven *Tetracladium* species have been reported from different continents [8,9], including *T. apiense* R.C. Sinclair & Eicker, *T. breve* A. Roldán, *T. ellipsoideum* M.M. Wang & Xing Z. Liu, *T. furcatum* Descals, *T. globosum* M.M. Wang & Xing Z. Liu, *T. marchalianum* De Wild., *T. maxilliforme* (Rostr.) Ingold, *T. nainitalense* (Sati & Arya), *T. palmatum* A. Roldán, *T. psychrophilum* M.M. Wang & Xing Z. Liu, and *T. setigerum* (Grove) Ingold [2,9–13]. The type species of the *Tetracladium* genus, *T. marchalianum*, was first reported in nature by de Wildeman [2]. Since Ingold described *T. setigerum* as an aquatic hyphomycete in 1942, the majority of *Tetracladium* species have been found in freshwater environments, although some have been found in terrestrial environments as well [14]. Most *Tetracladium* species produce tetracladiate conidia with several knobs

or finger-like processes that are a unique character distinguishing this genus from other aquatic hyphomycetes [15].

In this study, we isolated three *Tetracladium* species from freshwater environments in Korea. Through a molecular phylogenetic analysis and morphological characterization, the three fungal strains were identified as *T. fraxineum*, *T. setigerum,* and *T. marchalianum*. Here, we present mycological descriptions of the three *Tetracladium* species.

## 2. Materials and Methods

### 2.1. Isolation of Fungal Strains

We collected decaying leaf deposits and freshwater foam from streams. The decaying leaves were washed with distilled water (DW) and incubated in a pretreatment liquid medium (0.05% 3-morpholinopropane-1-sulfonic acid (weight/volume ($w/v$), 0.05% $KNO_3$ ($w/v$), 0.025% $KH_2PO_4$ ($w/v$), and 0.025% $K_2HPO_4$ ($w/v$)) at 20 °C overnight. The incubated supernatant, 100 μL, was spread on water agar (WA) (20 g/L) after 5 to 10 days. Hyphal tips and germinated conidia were detached under a microscope and transferred onto 24-well plates containing V8 agar (V8A) (8% V8 juice ($v/v$) and 1.5% agar ($w/v$) adjusted to pH 6.0 using 10 N NaOH). Then, the colonies were transferred onto potato dextrose agar (PDA) (Difco; BD, Franklin Lakes, NJ, USA) for pure culture. All strains used in this study were grown on PDA at 25 °C.

### 2.2. Induction of Sporulation and Investigation of Culture Conditions

For sporulation, the three *Tetracladium* species were incubated in sporulation medium (0.01% $CaCl_3 \cdot 2H_2O$ [$w/v$], 0.001% $MgSO_4 \cdot 7H_2O$ [$w/v$], 0.001% $KNO_3$ [$w/v$], 0.055% $K_2HPO_4$ [$w/v$], 0.05% MOPs buffer [$w/v$]) [16], at 20 °C, for 10 days. The strains were observed using a model Eclipse Ni-U microscope (Nikon, Tokyo, Japan).

To study the optimal growth conditions, the strains were cultivated on PDA, MEA (malt extract 2%; agar 2%/L), oatmeal agar (OA) ( Difco; BD, Franklin Lakes, NJ, USA), and yeast extract peptone dextrose agar (YPDA) (Duchefa Biochemie, Haarlem, The Netherlands), at 5, 15, 20, 25, 30, and 35 °C.

### 2.3. DNA Extraction and Phylogenetic Analysis

Fungal genomic DNA was isolated by means of a NucleoSpin Plant II DNA extraction kit (Macherey-Nagel, Duren, Germany). To identify the fungi, an internal transcribed spacer (ITS) region was amplified using primers ITS1 (5′-TCCGTAGGTGAACCTGCGG-3′) and ITS4 (5′-TCCTCCGCTTATTGATATGC-3′) [17]. DNA Homology searches were carried out by BLAST algorithms from the National Center for Biotechnology Information (NCBI). For the phylogenetic analysis, the MEGAX software was used [18]. A phylogenetic tree was constructed using a maximum parsimony (MP) analysis, the maximum likelihood (ML) method, and the neighbor-joining (NJ) method with 1000 bootstrap replications. The MP tree was obtained using the tree-bisection-regrafting (TBR) algorithm [19] with search level 1, in which the initial trees were obtained by the random addition of sequences (10 replicates). This analysis involved 36 nucleotide sequences. The final dataset contained 430 positions in total. The evolutionary distances were calculated using the Kimura 2-parameter method [20] and were in units of the number of base substitutions per site using the ML and NJ methods. The rate of variation among sites was modeled with a gamma distribution (shape parameter = 1). This analysis involved 36 nucleotide sequences. All positions with less than 95% site coverage were eliminated, i.e., fewer than 5% alignment gaps, missing data, or ambiguous bases were allowed at any position (partial deletion option). There was a total of 394 positions in the final dataset. Reference sequences of other fungi were obtained from GenBank at NCBI (Table 1).

**Table 1.** Reference sequences of strains from GenBank at the NCBI.

| Species | Strain | Accession Number |
| --- | --- | --- |
| | | ITS |
| *Dactylaria dimorphospora* | CBS 256.70 | U51980 |
| *Leohumicola minima* | N086 | HQ691252 |
| *Leohumicola verrucosa* | CBS 115881 | AY706323 |
| *Mycoarthris corallinus* | 91A | AF128440 |
| *Tetrachaetum elegans* | CB-M11 | KC834066 |
| *Tetracladium apiense* | CCM F-23199 | EU883420 |
| *Tetracladium apiense* | CCM F-23299 | EU883422 |
| *Tetracladium apiense* | CCM F-23399 | EU883421 |
| *Tetracladium breve* | CCM F-12505 | EU883431 |
| *Tetracladium ellipsoideum* | MIDUI20 | JX029111 |
| *Tetracladium ellipsoideum* | MIDUI21 | JX029124 |
| *Tetracladium ellipsoideum* | MIDUI30 | JX029113 |
| *Tetracladium fraxineum* sp. nov. | NNIBRFG814 | ON922528 |
| *Tetracladium furcatum* | C21BN1 | KT582084 |
| *Tetracladium furcatum* | CCM F-06983 | EU883428 |
| *Tetracladium furcatum* | CCM F-11883 | EU883432 |
| *Tetracladium furcatum* | GPO CO 01 | KC180667 |
| *Tetracladium globosum* | HAILUO215 | JX029109 |
| *Tetracladium globosum* | MY24 | JX029118 |
| *Tetracladium globosum* | MY25 | JX029133 |
| *Tetracladium marchalianum* | CCM F-11391 | EU883417 |
| *Tetracladium marchalianum* | CCM F-19399 | EU883423 |
| *Tetracladium marchalianum* | CCM F-26199 | AY204621 |
| *Tetracladium marchalianum* | CCM F-312 | EU883416 |
| *Tetracladium marchalianum* | NNIBRFG3891 | ON922530 |
| *Tetracladium maxilliforme* | CCM F-13186 | EU883430 |
| *Tetracladium maxilliforme* | CCM F-529 | EU883429 |
| *Tetracladium maxilliforme* | F-14286 | AF411027 |
| *Tetracladium psychrophilum* | HAILUO380 | JX029119 |
| *Tetracladium psychrophilum* | MY376 | JX029129 |
| *Tetracladium setigerum* | CCM F-10186 | EU883427 |
| *Tetracladium setigerum* | CCM F-19499 | EU883426 |
| *Tetracladium setigerum* | CCM F-20987 | KU519120 |
| *Tetracladium setigerum* | NNIBRFG1997 | ON922529 |
| *Tetracladium setigerum* | UMB 736.11 | KF952740 |
| *Tricladium alaskense* | VG-2012a | JQ417290 |

## 3. Results

### 3.1. Phylogeny

A BLASTn (Nucleotide BLAST) search of the ITS region of NNIBRFG814, NNI-BRFG1997, and NNIBRFG3891 revealed similarities of 100% (513/513), 98.96% (476/481), and 98.14% (475/484) with *T. breve* (MK371730), *T. setigerum* (KU519120), *T. marchalianum* (MK353126), respectively. A phylogenetic analysis was performed to identify three fungal strains, and this analysis involved 36 nucleotide sequences. There was a total of 394 positions in the final dataset. NNIBRFG1997 and NNIBRFG3891 were placed in clades with *T. setigerum* and *T. marchalianum*, respectively. NNIBRFG814 was placed in a clade with *T. breve* on the MP and NJ trees, but not in any clade on the ML tree. The estimated value of the shape parameter for the discrete gamma distribution was 0.4286. Substitution patterns and rates were estimated under the Kimura [20] model (+G). The nucleotide frequencies were A = 25.00%, T/U = 25.00%, C = 25.00%, and G = 25.00%. The maximum log-likelihood for this computation was −2584.030.

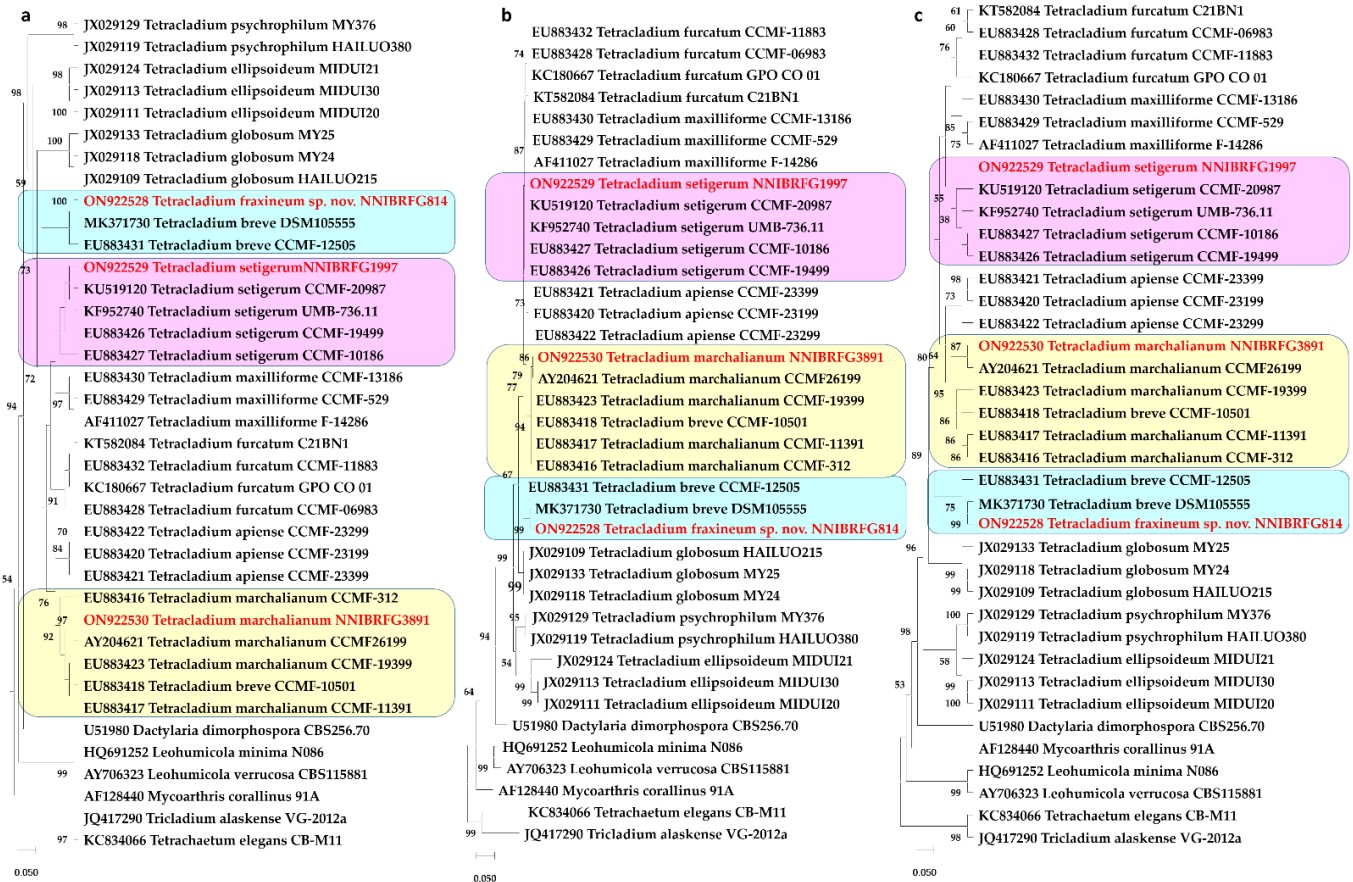

**Figure 1.** Phylogenetic analysis using: (**a**) the maximum parsimony method; (**b**) maximum likelihood method; (**c**) neighbor-joining method, based on the ITS region of three *Tetracladium* species. This analysis involved 36 nucleotide sequences. *Tetrachaetum* elegans and *Tricladium alaskense* were used as outgroups. At branches, bootstrap values of more than 50% (1000 replications) were shown. The unrecorded strains are shown in bold and red.

### 3.2. Taxonomy

#### 3.2.1. *Tetracladium fraxineum* sp. nov. J. Goh & H.Y. Mun (2022)

Etymology: Referring to the host genus *Fraxinus*

Mycobank No.: MB827330

Description: Colonies on PDA, with a diameter of 58–60 mm after 14 days at 20 °C, covered by aerial mycelia, flat floccose white to pale pink, floccose white concentric ring, margin regular, reverse white, and apricot near the center. Colonies on OA, with a diameter of 40–46 mm after 14 days at 20 °C, covered by cottony aerial mycelia, flat floccose pale pink concentric ring, white margin regular, and reverse apricot. Colonies on MEA, with a diameter of 56–58 mm after 14 days at 20 °C, covered by white floccose aerial mycelia, a white concentric ring, margin filiform, reverse white, and a light yellow near the center. Colonies on YPDA, with a diameter of 16–18 mm after 14 days at 20 °C, flat floccose aerial mycelia, umbonate, margin undulate, dense, front, and reverse apricot. Sporulation when submerged. Conidia were three digitiform 20–32 μm and short 1–3 septate, distal cell globose to ellipsoid 6–8 μm long × 4–5 μm width, primary branches at one level, immediately below the globose cell, typically 1-septate, acicular secondary branches were inserted below the second globose cell, and straight and slightly curved branches (Figures 1–3).

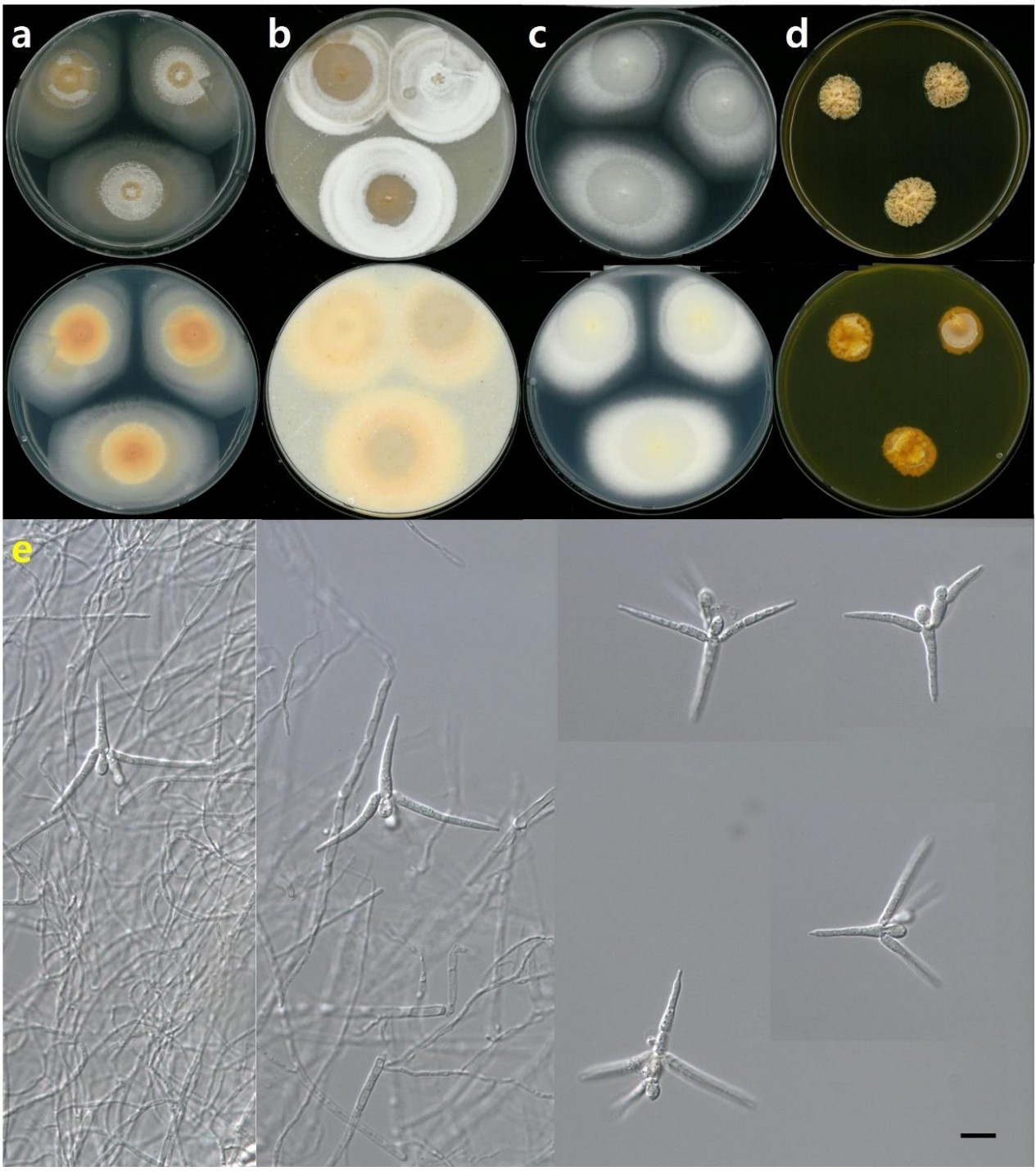

**Figure 2.** Colony shape and conidia of *Tetracladium fraxineum* NNIBRFG814. Colony shape of front and back on: (**a**) PDA; (**b**) OA; (**c**) MEA; (**d**) YPDA; (**e**) conidia, on SM media, 400× (size bar = 10 μm).

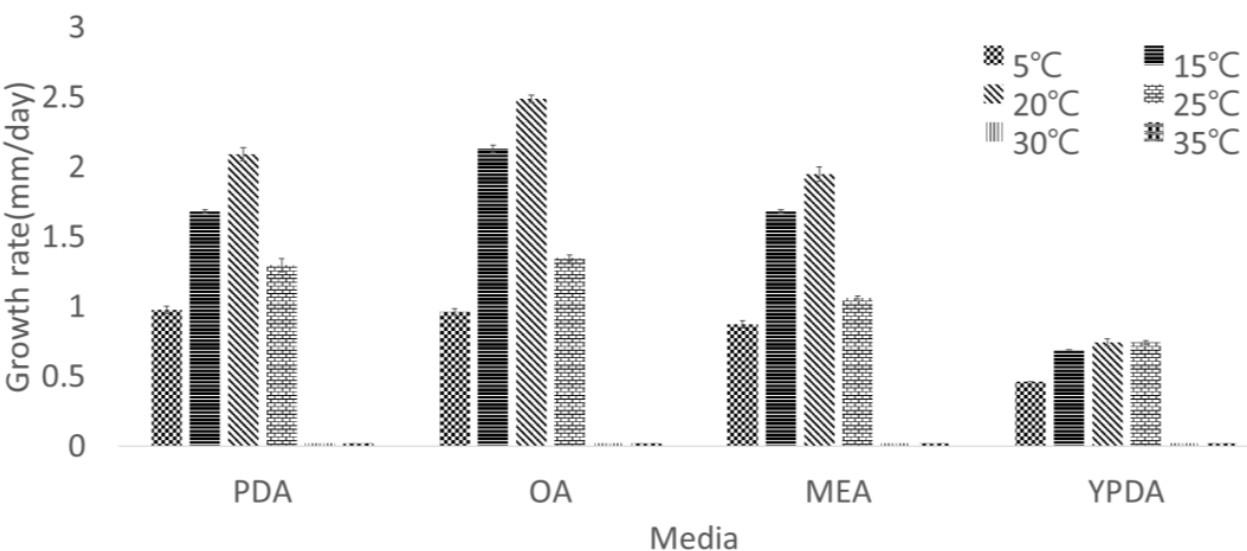

**Figure 3.** The growth rate of NNIBRFG814 depending on four media and six temperature.

Habitat: Decaying leaves of *Fraxinus rhynchophylla* from freshwater

Specimen examined: Taebaek-si, Gangwon-do, Republic of Korea, 3 February 2016, NNI-BRFG814, Nakdonggang National Institute of Biological Resources (Sangju-si, Gyeongsangbuk-do, Republic of Korea)

Note: Based on the MP and NJ trees, this strain was in a clade with *T. breve*. Sequence similarity to *T. breve* was very high, 100% (MK371730) (Figure 1), but morphology was similar to *T. marchlianum*. Conidia were found which were very similar to those of *T. marchalianum* but consistently much smaller and shorter (Figure 2). Therefore, NNIBRFG814 was named *T. fraxineum*, as a new species.

NNIBRFG814 grew very slowly, approximately 2.5 mm/day, which was the best growth on OA at 20 °C. NNIBRFG814 grew at 5 °C as a psychrophilic fungus, but did not grow at 30 °C (Figure 3).

### 3.2.2. *Tetracladium setigerum* (Grove) Ingold

Basionym: *Tetracladium setigerum* (Grove) Ingold, Transactions of the British Mycological Society 25 (4): 369 (1942)

Mycobank No.: MB291361

Description: Colonies on PDA were 14–18 mm in diameter after 14 days at 20 °C, covered by flat floccose aerial mycelia, yellow, irregular sulcate, margin undulate, umbonate elevation, and reverse yellow. Colonies on OA, 25–34 mm after 14 days at 20 °C, covered by floccose aerial mycelia, white to yellow, with a white concentric ring, margin regular and reverse apricot. Colonies on MEA, 33–36 mm after 14 days at 20 °C, covered by aerial mycelia, white to hyaline, white concentric ring, and reverse white to hyaline, white near the center. Colonies on YPDA, after 14 days at 20 °C, covered by hyaline aerial mycelia. The strain was not formed into a colony but grew into hyaline mycelia. Sporulation when submerged. Conidiophores were semi-macronematous and typically simple. Conidia were two or three digitiforms, 30–80 μm long, and three narrow obclavate elements, 12–20 μm long and 4–8 μm wide, with the upper digitiform having primary branches arising at two levels, curved adaxially, 3 or 5 septate, long and pointed branches (Figures 1 and 4).

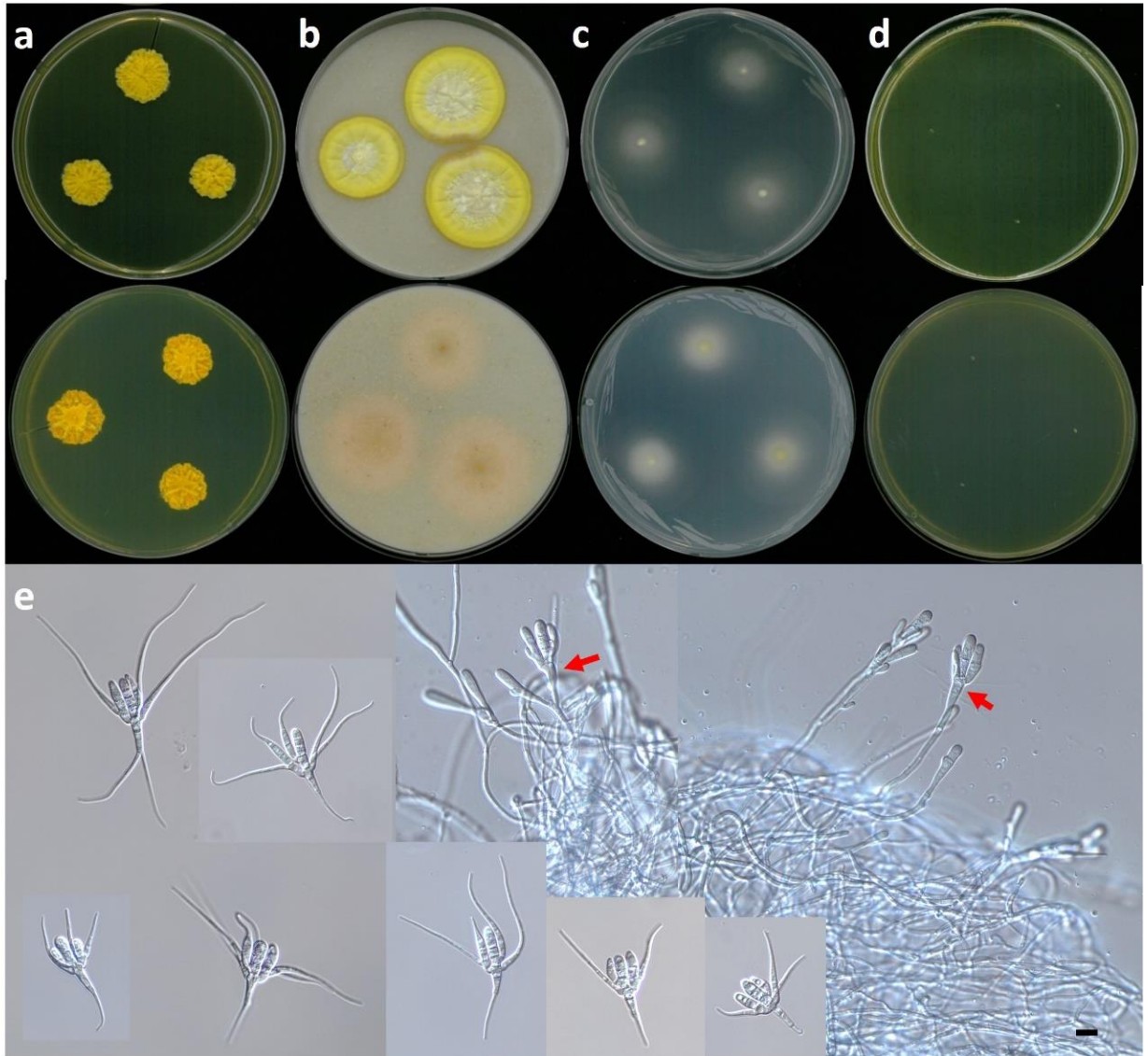

**Figure 4.** Colony shape and conidia of *Tetracladium setigerum* NNIBRFG1997 Colony shape of front and back on: (**a**) PDA; (**b**) OA; (**c**) MEA; (**d**) YPDA; (**e**) conidia, on SM media, 400× (size bar = 10 μm). Red arrows point at conidiophores.

Habitat: Freshwater foam in a stream

Specimen examined: Gam cheon, Gimcheon-si, Gyeonsangbuk-do, Republic of Korea, 23 March 2016, NNIBRFG1997, Nakdonggang National Institute of Biological Resources (Sangju-si, Gyeongsangbuk-do, Republic of Korea)

Note: Based on the phylogenetic tree, this strain was in the *T. setigerum* clade. Similarity was very low, i.e., 98.96% (KU519120), but the morphology was similar to *T. setigerum.* As a result, NNIBRFG1997 was considered to be *T. setigerum*.

### 3.2.3. *Tetracladium marchalianum* De Wild

Basionym: *Tetracladium marchalianum* De Wild., Annales de la Societé Belge de Microscopie 17: 39 (1893)

Mycobank No.: MB243554

Description: Colonies on PDA, 45–47 mm in diameter after 14 days at 20 °C, covered by floccose aerial mycelia, white to pale apricot, pale apricot concentric ring, irregular sulcate, white margin filiform, and reverse apricot. After 14 days at 20 °C, colonies on OA, 37–43 mm in diameter, were covered by floccose aerial mycelia, pale yellow mixed white,

pale yellow concentric ring, regular margin, and reverse mustard. Colonies on MEA were 28–35 mm in diameter after 14 days at 20 °C, covered by flat floccose near the center, pale apricot to white, floccose margin filiform, and reverse pale apricot. Colonies on YPDA were 30–32 mm in diameter after 14 days at 20 °C, floccose aerial mycelia, light yellow to pale yellow, irregular sulcate, and reverse mustard. Sporulation when submerged. Conidia had an axis of 1–3 septate and a length of 15–25 μm, three or four digitiforms, 1–3 septate, and a length of 25–40 μm. The distal cell was bicellular and constricted at the septum. Globose to ellipsoid was 9–10 μm long and 4–6 μm wide, primary branches at one level were immediately below the globose cell, acicular secondary branch was inserted below the second globose cell, and branches were a little curved (Figures 1 and 5).

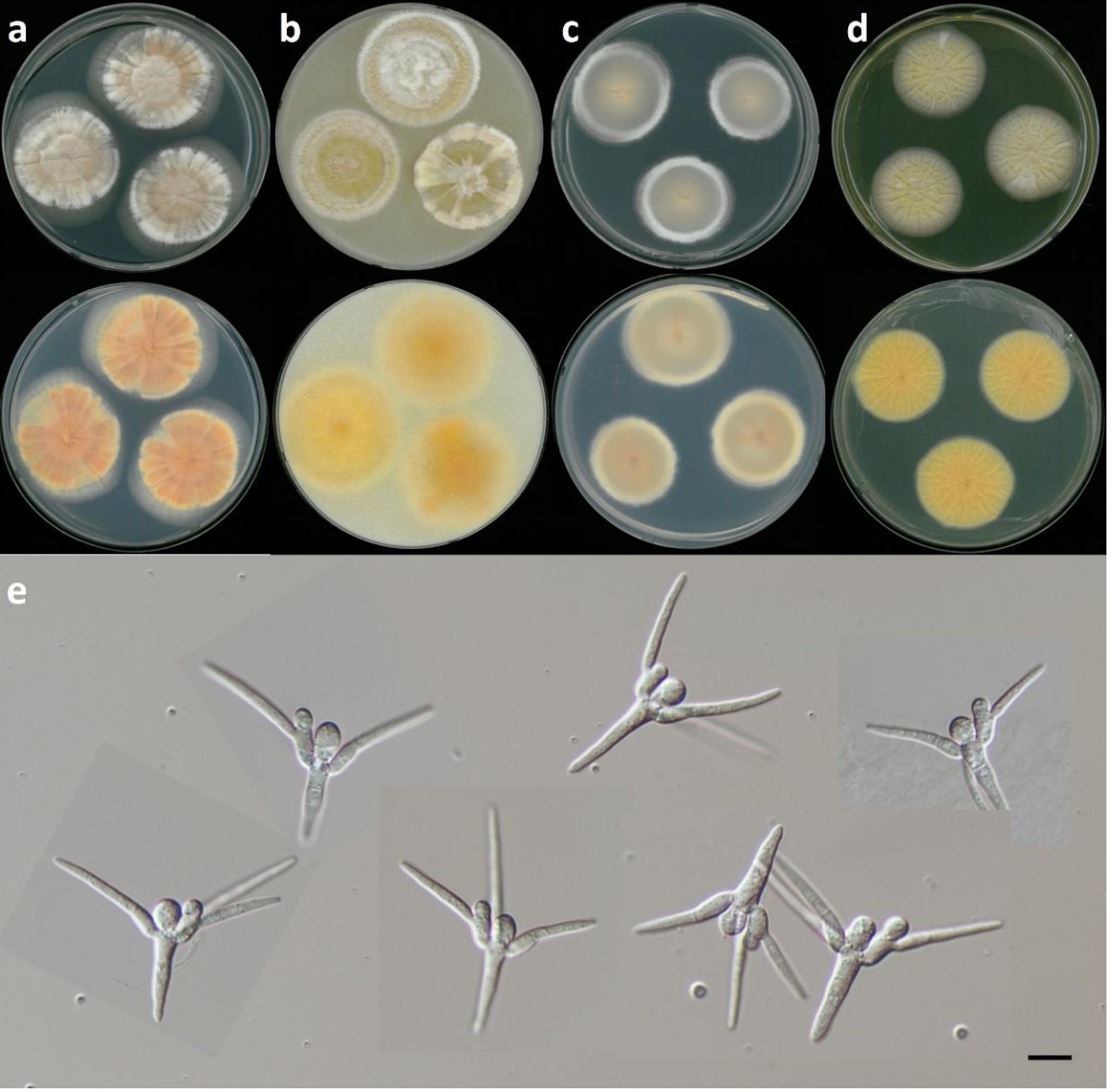

**Figure 5.** Colony shape and conidia of *Tetracladium marchalianum* NNIBRFG3891: Colony shape of front and back on: (**a**) PDA; (**b**) OA; (**c**) MEA; (**d**) YPDA; (**e**) conidia, on SM media, 400× (size bar = 10 μm).

Habitat: Filtered freshwater in stream

Specimen examined: Yangsan cheon, Mungyeon-si, Gyeonsangbuk-do, Republic of Korea, 24 February 2017, NNIBRFG3891, Nakdonggang National Institute of Biological Resources (Sangju-si, Gyeongsangbuk-do, Republic of Korea)

Note: Based on the phylogenetic tree, this strain was in the *T. marchalianum* clade. Similarity was very low, i.e., 98.14% (MK353126), but morphology was similar to *T. marchlianum*. As a result, NNIBRFG3891 was considered to be *T. marchalianum*.

## 4. Discussion

The morphology of *T. breve* was three long and thin digitiform elements, two narrow obclavate, filiform elements on a stalk, similar to *T. setigerum* [15]. In contrast, NNIBRFG814 had three short digitiform, globose distal cells and two level branches similar to those of *T. marchalianum*. *T. setigerum* was similar to *T. breve*, but it had three narrow obclavate elements [15]. The morphology of NNIBRFG1997 included three narrow obclavate elements, similar to *T. setigerum*. Three species, *T. fraxineum* NNIBRFG814, *T. stigerum* NNIBRFG1997, and *T. marchalianum* NNIBRFG 3891, could not produce conidia on solid media, but could produce conidia and conidiophores in liquid media at 20 °C. Almost Ingoldian fungi could not produce conidia in solid media, but they could form conidia when they were completely submerged in water [2].

We collected *Tetracladium* species from decaying leaves, freshwater foam, and filtered freshwater in late winter. Wang et al. studied cold-adapted fungi and reported three *Tetracladium* species, which grew well at temperatures below 20 °C. They collected *Tetracladium* species from soil on a glacier [13]. Our strains were collected from freshwater, in cold weather in Feburary and March, at temperatures under 0 °C. Our three strains grew well below 20 °C as psychrophilic fungi. Czeczuga and Orlowska investigated the aquatic hyphomycetes diversity in melting snow water, rain, and ice. According to them, as the water temperature decreased, the increasing surface tension of the upper layer seemed to pull various aquatic fungi. They found most species in their study from melting snow water [21]. For effective isolation of aquatic hyphomycetes including *Tetracladium*, we would consider sampling in freshwaters during cold weather.

**Author Contributions:** H.Y.M., conceptualization, identification, illustrations editing, writing the orginal draft, editing, and correspondence; J.G., conceptualization, collection, identification, editing, assistance in reviewing, funding; Y.J.J., methodology, illustrations editing. All authors have read and agreed to the published version of the manuscript.

**Funding:** This work was supported by a grant from the Nakdonggang National Institute of Biological Resources (NNIBR), funded by the Ministry of Environment (MOE) of the Republic of Korea (NNIBR202201107).

**Institutional Review Board Statement:** Not applicable.

**Data Availability Statement:** The types were deposited in the Fungal research team, Nakdonggang National Institute of Biological Resources, Korea.

**Conflicts of Interest:** The authors declare no conflict of interest.

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
