# Peer review of "New Records of the Psychrophilic Tetracladium Species Isolated from Freshwater Environments in Korea"

_diversity, doi:10.3390/d14100789_

Round 1

Reviewer 1 Report

My comments on the manuscript are marked in the body of the manuscript. The manuscript needs to be revised in the light of comments given. There are some confusions in the manuscript, which needs to be clarified. The results are not discussed properly and needs improvement.     

Author Response

Thank you for your kind comment.

I checked your opinion, and then I  modified the content.

I tried to discuss properly and improvement.

If you have good opinions, recommend, Please.

Reviewer 2 Report

The manuscript presents and original study concerning some new aquatic fungi from Tetracladium genus. All the sections of the manuscript are well written and presented.

Some small corrections should be done:

Line 63: was, not were

Line 64: detached, not isolated

Lines 70, 71, 74, 115 and elsewhere: remove spaces between the value (number) and symbol “%”

Line 76, 130 and elsewhere: put a space between value and “°C”

Lines 194-195: apricot, not aricot.

Author Response

(The authors gave the same response as above.)

Round 2

Reviewer 1 Report

The manuscript is now in a presentable form and can be accepted.